# Autosomal recessive hyposegmentation of granulocytes in Australian Shepherd Dogs indicates a role for *LMBR1L* in myeloid leukocytes

Bianca Lourdes Frehner[1☯], Matthias Christen[2☯], Iris M. Reichler[1],
Vidhya Jagannathan[2], Marilisa Novacco[3], Barbara Riond[3], Laureen M. Peters[4],
José Suárez Sánchez-Andrade[5], Aldona Pieńkowska-Schelling[2,6], Claude Schelling[6],
Anja Kipar[7], Tosso Leeb[2‡*], Orsolya Balogh[1,8‡]

1 Clinic of Reproductive Medicine, Vetsuisse Faculty, University of Zurich, Zurich, Switzerland, 2 Institute of Genetics, Vetsuisse Faculty, University of Bern, Bern, Switzerland, 3 Clinical Laboratory, Vetsuisse Faculty, University of Zurich, Zurich, Switzerland, 4 Clinical Diagnostic Laboratory, Vetsuisse Faculty, University of Bern, Bern, Switzerland, 5 Clinic for Diagnostic Imaging, Vetsuisse Faculty, University of Zurich, Zurich, Switzerland, 6 Research Platform AgroVet-Strickhof, Vetsuisse Faculty Zurich, University of Zurich, Lindau, Switzerland, 7 Institute of Veterinary Pathology, Vetsuisse Faculty, University of Zurich, Zurich, Switzerland, 8 Department of Small Animal Clinical Sciences, Virginia-Maryland College of Veterinary Medicine, Blacksburg, Virginia, United States of America

☯ These authors contributed equally to this work.
‡ TL and OB also contributed equally and share senior authorship to this work.
* tosso.leeb@unibe.ch

**Data Availability Statement:** All data are freely available. The reported whole genome sequence data have been deposited in the European

## Abstract

Pelger-Huët anomaly (PHA) in humans is an autosomal dominant hematological phenotype without major clinical consequences. PHA involves a characteristic hyposegmentation of granulocytes (HG). Human PHA is caused by heterozygous loss of function variants in the *LBR* gene encoding lamin receptor B. Bi-allelic variants and complete deficiency of *LBR* cause the much more severe Greenberg skeletal dysplasia which is lethal in utero and characterized by massive skeletal malformation and gross fetal hydrops. HG phenotypes have also been described in domestic animals and homology to human PHA has been claimed in the literature. We studied a litter of Australian Shepherd Dogs with four stillborn puppies in which both parents had an HG phenotype. Linkage analysis excluded *LBR* as responsible gene for the stillborn puppies. We then investigated the HG phenotype in Australian Shepherd Dogs independently of the prenatal lethality. Genome-wide association mapped the HG locus to chromosome 27 and established an autosomal recessive mode of inheritance. Whole genome sequencing identified a splice site variant in *LMBR1L*, c.191+1G>A, as most likely causal variant for the HG phenotype. The mutant allele abrogates the expression of the longer X2 isoform but does not affect transcripts encoding the shorter X1 isoform of the LMBR1L protein. The homozygous mutant *LMBR1L* genotype associated with HG is common in Australian Shepherd Dogs and was found in 39 of 300 genotyped dogs (13%). Our results point to a previously unsuspected function of *LMBR1L* in the myeloid lineage of leukocytes.

Nucleotide Archive. Accessions are given in the S2 Table.

**Funding:** This study was funded by grant no. 138 from the Albert Heim Foundation to I.M.R. and O.B. The funders had no role in study design, data collection and analysis, decision to publish, or preparation of the manuscript.

**Competing interests:** The authors declare no conflicts of interests.

## Author summary

Pelger-Huët anomaly (PHA) is a clinically benign hematological phenotype characterized by hyposegmentation of the nuclei of granulocytes. Human PHA is caused by heterozygous loss of function variants in the *LBR* gene encoding lamin receptor B. Bi-allelic *LBR* loss of function causes Greenberg skeletal dysplasia which results in fetal lethality. Hyposegmentation of granulocytes (HG) is common in the Australian Shepherd Dog breed and has been claimed to represent a true homolog of human PHA, resulting in breeding restrictions for affected dogs. In this study, we provide evidence that HG in Australian Shepherd Dogs is an autosomal recessive trait without major negative consequences for health. We identify an *LMBR1L* splice site variant as likely causal variant. LMBR1L is a scarcely characterized protein that so far has been associated with lymphopoiesis in mice. The identified *LMBR1L* variant in HG affected Australian Shepherd Dogs selectively abrogates the expression of the presumably full-length transcript isoform. Homozygous mutant dogs still express an alternative shorter transcript, which encodes an N-terminally truncated protein isoform. Our results indicate a previously unsuspected functional role of LMBR1L in myeloid cells.

## Introduction

Pelger-Huët anomaly (PHA) is a benign congenital condition in humans characterized by hyposegmentation and coarse chromatin structure of polymorphonuclear white blood cells [1]. The human PHA phenotype is inherited in an autosomal dominant manner [2–4]. It is caused by heterozygous variants in the *LBR* gene, coding for the lamin B receptor, a 70.4-kDa protein of the inner nuclear membrane that contains an N-terminal lamin B/DNA-binding domain of $\sim$200 amino acids followed by a C-terminal sterol reductase-like domain of $\sim$450 amino acids, which exhibits sterol Δ14-reductase activity (OMIM #600024) [5–7]. Complete loss of function due to bi-allelic *LBR* variants causes the severe Greenberg skeletal dysplasia, a lethal chondrodystrophy characterized by fetal hydrops, short limbs, and abnormal chondro-osseus calcification [7,8].

The murine *Lbr* gene corresponds to the ichthyosis (*ic*) locus in mice. Heterozygous *Lbr*$^{+/-}$ mice exhibit abnormalities in nuclear heterochromatin similar to those observed in PHA, whereas *Lbr*$^{-/-}$ animals develop ichthyosis [9]. Another *Lbr*$^{-/-}$ mouse strain shows embryonic lethality with incomplete penetrance, hydrocephalus and syndactyly, similar to human cases of Greenberg skeletal dysplasia [10].

In domestic animals, PHA has been reported in rabbits [11–13], cats [14–16], horses [17,18], and different dog breeds [19–26]. Autosomal dominant inheritance of PHA has been reported in rabbits [12,13] and one litter of cats [15]. PHA homozygosity in rabbits and cats has been described as a predominance of round or oval granulocyte nuclei with clumped chromatin, together with highly variable skeletal anomalies including shortened long bones and ribs [12,13,16]. To the best of our knowledge, the molecular basis for PHA has not yet been reported in any domestic animal species.

In a study involving 892 Australian Shepherd Dogs including 87 PHA cases, an autosomal dominant mode of inheritance with incomplete penetrance was suggested [24]. This study on Australian Shepherd Dogs noticed the complete absence of the hematological changes typically associated with the homozygous state of PHA, leading the authors to conclude that "the homozygous form of the anomaly is lethal in utero" [24].

Our investigation was prompted by the report of a litter of Australian Shepherd Dogs from two parents with hyposegmentation of granulocytes (HG), in which four out of seven offspring were stillborn. Based on the available literature we initially hypothesized that these stillborn puppies represented homozygotes of a canine form of PHA. As the obtained results clearly refuted this hypothesis, we further investigated the genetic basis of the HG phenotype in the Australian Shepherd Dog population.

## Results

### Phenotypic prevalence of HG in Australian Shepherd Dogs

We investigated the prevalence of the HG phenotype in 77 clinically healthy Australian Shepherd Dogs from Switzerland (S1 Table). The age of the dogs ranged from 6 months to 13.2 years, with a median age of 3.6 years. Eleven of the 77 dogs were HG affected resulting in a prevalence of 14% in this cohort. No sex predilection was observed for HG (five female and six male).

### Clinical investigations in a family with stillborn puppies

A female Australian Shepherd Dog was presented for pregnancy check at mid-gestation. During ultrasound examination, 6–7 vital fetuses and two embryonic resorption sites were identified. This prompted further diagnostic tests, which then revealed the HG phenotype (Fig 1).

The bitch whelped naturally three viable and four stillborn puppies. In all stillborn puppies, post mortem whole-body CT scans revealed moderate to severe ex-vacuo hydrocephalus and normal musculoskeletal structures, i.e. normal length and shape of all bones of the appendicular and axial skeleton without signs of physeal abnormalities or growth plate disturbance. The gross post mortem findings were restricted to the head and brain. The skulls showed mild calvarial bulging and a mild dome shaped deformation as well as a small, non-protruding persistent fontanelle. The cerebral hemispheres were represented by membranous, fluid-filled sacs, suggesting severe atrophy of the parenchyma. The cerebellum was reduced in size and displayed an unstructured surface (cerebellar hypoplasia). All other bones did not display any gross abnormalities. The histological examination performed on one puppy of the cerebrum showed diffuse malacia of the remaining parenchyma, multifocal dystrophic mineralisation

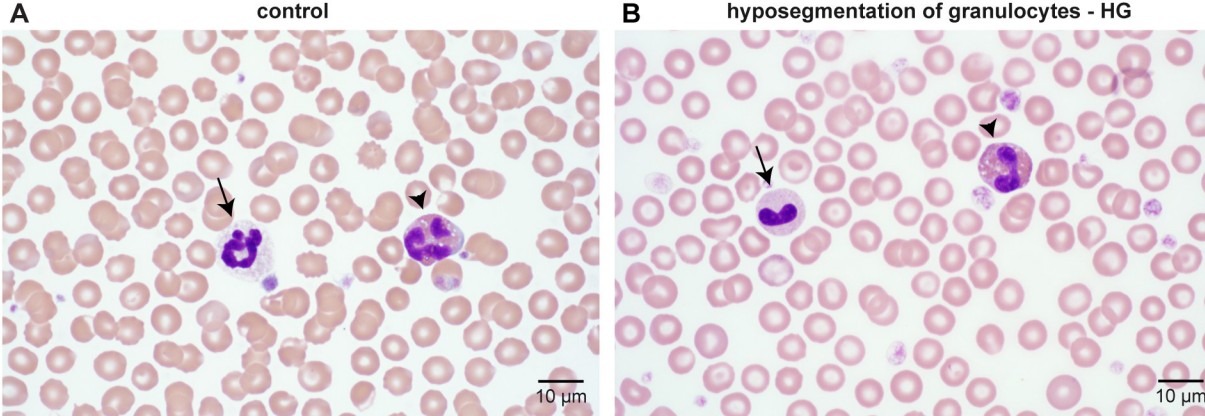

**Fig 1. HG phenotype.** (**A**) Photomicrograph of a peripheral blood smear of a control dog showing normal nuclear segmentation of a neutrophil granulocyte (arrow) and an eosinophil granulocyte (arrowhead). (**B**) Peripheral blood smear of the pregnant dam of the litter showing a lack of nuclear segmentation in a neutrophil granulocyte (arrow) and an eosinophil granulocyte (arrowhead).

and replacement of neurons by Gitter cells. The ventricular structures were lined by a discontinuous layer of ependymal cells. The thoracic vertebrae, ribs, sternum, femur and joint-forming bones of the elbow and shoulder joint were processed for the histological examination and examined. None of these exhibited any histological abnormality. The epiphyseal plates showed physiological cartilage zones, and the ossification process appeared unaltered.

Additionally, a HG phenotype was confirmed in the sire and the three surviving offspring. Due to these results and based on the available literature, we initially hypothesized that the stillborn puppies resembled a more severe phenotype reminiscent of PHA homozygosity while the parents and surviving puppies with HG represented heterozygous animals with PHA.

### Genetic analysis of the litter with stillborn puppies

We performed parametric linkage analysis in the family with the four stillborn puppies. Parameters were set for autosomal recessive inheritance of the lethality phenotype. This linkage analysis yielded positive LOD scores in 6 segments of the genome comprising a total of 9.3 Mb or 0.4% of the genome (S3 and S4 Tables). Only 900 kb on chromosome 21 showed evidence of combined linkage and homozygosity in the four stillborn puppies (S5 and S6 Tables). The entire chromosome 7, harboring the *LBR* gene associated with the human PHA phenotype, had LOD scores < -4 and was unambiguously excluded in our linkage analysis.

We additionally sequenced the genome of one of the stillborn puppies. However, this analysis did not reveal any obvious plausible candidate causative variants for the prenatal lethality (S7 Table).

### Establishing the mode of inheritance of HG

As our data were incompatible with a dominant mode of inheritance for the HG phenotype, we revisited the previously reported data on 892 Australian Shepherd Dogs [24]. Latimer and colleagues hypothesized that the HG phenotype in Australian Shepherd Dogs represented the canine homolog of human PHA and analyzed their data under a model of autosomal dominant inheritance. The authors noted that the expected incidence of HG in F1 progeny of parents with known HG phenotypes deviated from the expected values for a dominant trait. They further described that the phenotype sometimes skipped generations and reported six HG affected puppies that were born in four different litters from unaffected parents. Latimer and colleagues briefly discussed the alternative hypothesis of autosomal recessive inheritance and stated that "support for this mode of inheritance is present but tenuous" [24]. Ultimately, the authors rejected autosomal recessive inheritance and concluded from their data that HG in Australian Shepherd Dogs follows autosomal dominant inheritance with incomplete penetrance [24].

We re-analyzed the incidence data from [24] in F1 offspring from parents with known phenotypes. We considered three different alternative hypotheses for the mode of inheritance: autosomal recessive, autosomal dominant and autosomal semi-dominant with prenatal lethality for the homozygous mutant F1 offspring. The previously reported incidence data were incompatible with the dominant or the semi-dominant model, but they fit perfectly to a fully penetrant autosomal recessive mode of inheritance for HG (S1 Fig).

### Mapping of the HG locus

We then performed a GWAS for HG with 76 Australian Shepherd Dogs. After quality control, the pruned dataset consisted of 22 cases, 54 controls, and 79,630 markers. A single association signal with three markers exceeding the Bonferroni corrected significance threshold of $p = 6.27 \times 10^{-7}$ was obtained. All associated markers were located on chromosome 27 within

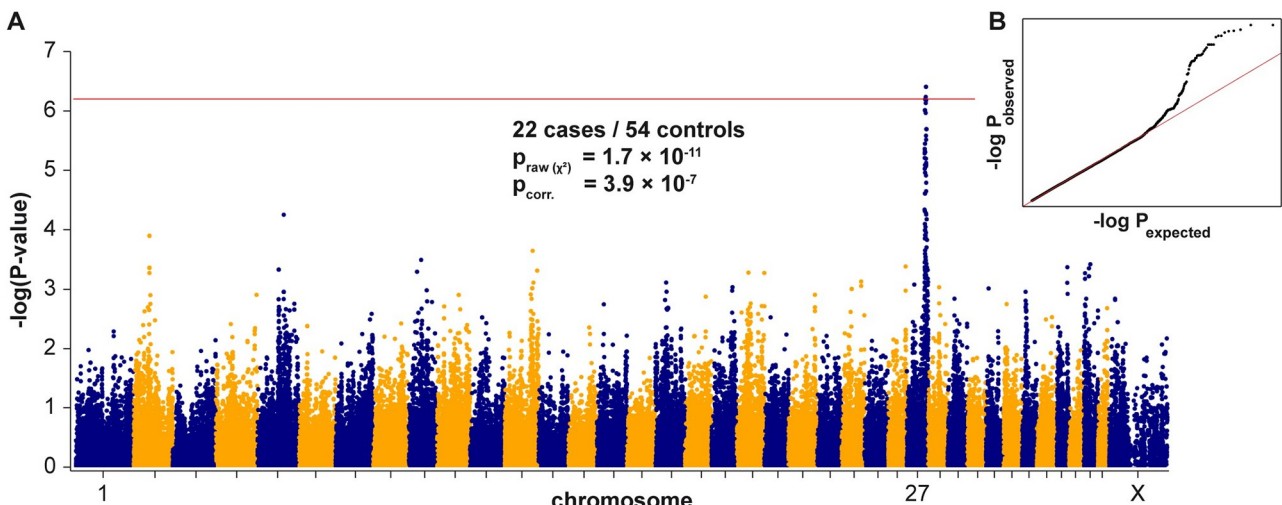

**Fig 2. Mapping of the HG locus by genome-wide association.** (**A**) Manhattan plot illustrating a single signal on chromosome 27. The red line indicates the Bonferroni threshold for significance at $p = 6.27 \times 10^{-7}$. The genomic inflation factor was 0.99. (**B**) The quantile–quantile (QQ) plot shows the observed versus expected -log(p) values. The straight red line in the QQ plot indicates the distribution of p-values under the null hypothesis. The deviation of p-values above the line indicates that these markers are stronger associated with the trait than would be expected by chance. This supports the biological significance of the association.

an interval spanning from 41.1 Mb—41.6 Mb. The top-associated marker at Chr27:41,593,807 had a p-value of $3.9 \times 10^{-7}$ (Fig 2). The genotype distribution at the associated markers confirmed the autosomal recessive inheritance of the HG phenotype (S8 Table).

In order to fine-map the HG locus, we performed autozygosity mapping. All 22 cases from the GWAS shared a 281 kb region with identical homozygous haplotypes that overlapped the GWAS signal (S9 Table). The critical interval for the HG locus corresponded to the interval between the first flanking heterozygous markers on either side of the homozygous segment or Chr27:40,999,575–41,285,438 (UU_Cfam_GSD_1.0 assembly).

### Identification of a candidate causative variant

Whole genome sequence data of one of the stillborn puppies and another HG affected Australian Shepherd Dog were analyzed. The variants in these two genomes were compared to genome sequence data of 918 control dogs of different breeds (S10 Table). Three private variants were identified in the critical interval in the affected dogs, and only one of these variants was predicted to be protein changing (Table 1). A visual inspection of the short-read alignments did not reveal any additional structural variants affecting protein-coding sequences in the critical interval.

**Table 1. Variant filtering in two genomes of HG affected dogs against 918 control genomes.**

| Filtering step | Homozygous variants |
| --- | --- |
| Shared variants in whole genome | 1,348,510 |
| Shared variants, private to cases, in whole genome | 77 |
| Shared variants in critical interval | 215 |
| Shared variants, private to cases, in critical interval | 3 |
| Protein changing variants, private to cases, in critical interval | 1 |

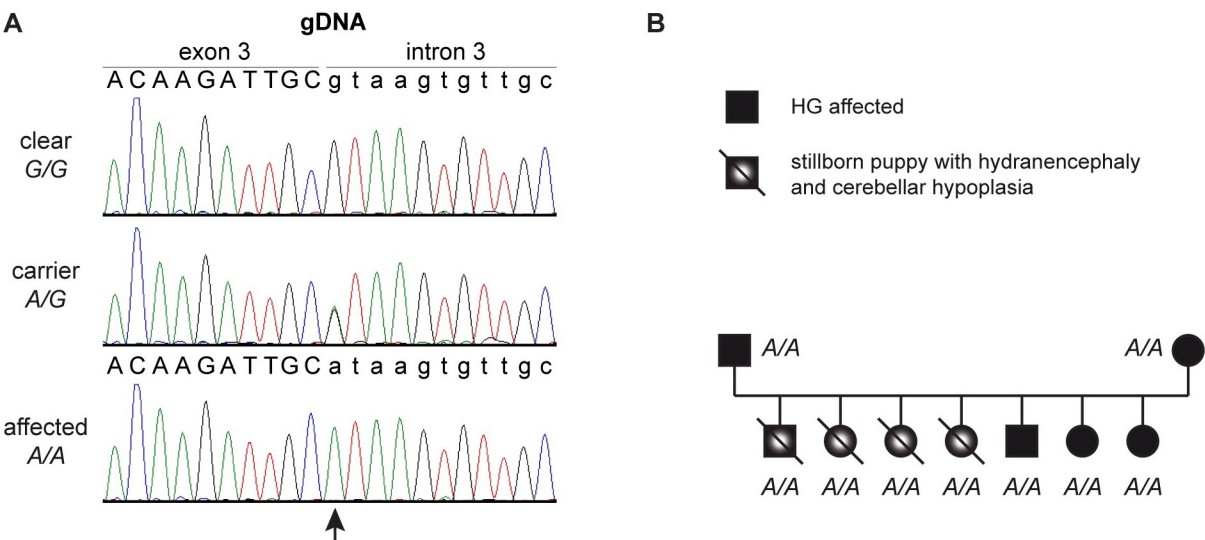

**Fig 3. Genotyping of the *LMBR1L*:c.191+1G>A splice site variant.** (**A**) Representative Sanger electropherograms from dogs with three different genotypes are shown. The base exchange is indicated with an arrow. (**B**) All members of the family with the four stillborn puppies were homozygous for the mutant *A/A* genotype.

The remaining variant, Chr27:41,169,674C>T, affected a splice site of the *LMBR1L* gene, XM_038577534.1:c.191+1G>A. The presence of the variant was confirmed by Sanger sequencing (Fig 3).

We genotyped 300 Australian Shepherd Dogs for the *LMBR1L*:c.191+1G>A variant. The cohort included the dogs used for the initial GWAS. We observed perfect genotype-phenotype association in 26 HG cases and 68 HG controls (Table 2).

The remaining 206 Australian Shepherd Dogs represented a convenience sample from the breed, in which the HG phenotypes were unknown at the time of genotyping. The mutant allele frequency in these dogs was 15.0%. The genotype distribution in the 206 dogs deviated from Hardy-Weinberg equilibrium with an observed excess of homozygous mutant dogs ($p = 6.0 \times 10^{-6}$), most likely due to non-representative sampling.

## Functional confirmation at the transcript level

According to NCBI annotation 106, the canine *LMBR1L* gene gives rise to three different alternative transcripts encoding two different proteins termed isoform X1 (short, 362 aa) and X2 (long, 489 aa). On the mRNA level, the main difference is the inclusion or exclusion of exon 3 in the transcripts encoding these two protein isoforms. Due to the utilization of different start codons in the transcripts, the X2 protein isoform contains an additional 127 amino acids at the N-terminus compared to the shorter X1 isoform.

**Table 2. Association of the genotypes at the *LMBR1L*:c.191+1G>A variant in 300 Australian Shepherd Dogs.**

| HG phenotype | G/G | G/A | A/A |
|---|---|---|---|
| Cases (n = 26) | - | - | 26 |
| Controls (n = 68) | 43 | 25 | - |
| Dogs with unknown HG phenotype (n = 206) | 157 | 36 | 13 |

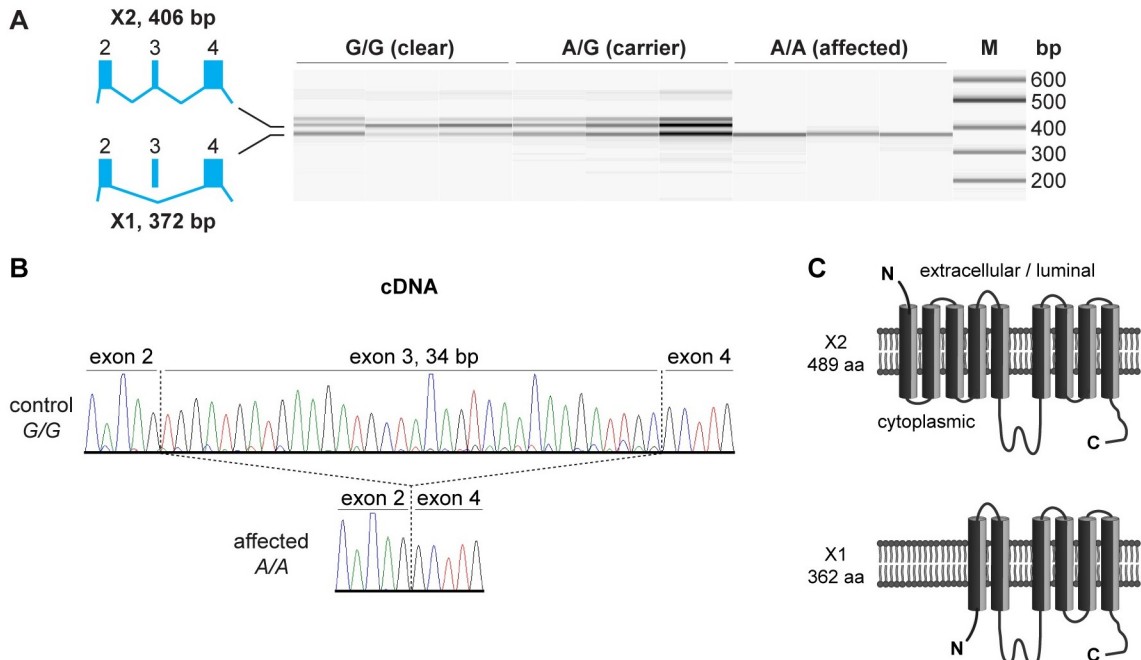

**Fig 4. Effect of the *LMBR1L*:c.191+1G>A variant on splicing.** (**A**) RT-PCR analysis of *LMBR1L* transcripts in three dogs of each genotype. Dogs with at least one copy of the wildtype allele express transcripts encoding the X1 and X2 isoforms of the LMBR1L protein. Homozygous mutant dogs only express transcripts encoding the shorter X1 isoform. The sizes of the two lower bands differ by 34 bp corresponding to the presence or absence of exon 3 in the *LMBR1L* transcripts. The third and largest band visible in the control dogs most likely represents heteroduplex molecules consisting of one strand of the short transcript annealed to one strand of the long transcript. (**B**) Sanger sequencing of the RT-PCR products confirmed the splice junctions. Please note that these sequences were obtained with a reverse primer. In the control animal, the primary major signals correspond to the long transcript including exon 3. Overlapping small secondary signals corresponding to the shorter transcript can be seen on the left side of exon 4. The signal ratio indicates that the longer transcript isoform is much more abundant than the shorter isoform in control dogs. (**C**) Predicted membrane topology of the long and short LMBR1L isoform. The long isoform has 9 transmembrane domains and is expressed in the ER and plasma membrane [27]. The shorter X1 isoform is predicted to lack 127 aa at the N-terminus comprising the first two and a large part of the third transmembrane domain.

As the genomic variant affected the canonical GT-dinucleotide at the 5'-splice site of intron 3, we experimentally assessed the consequences of the base substitution on the transcript levels in three dogs of each genotype. All experiments were performed with RNA from blood cells.

Homozygous mutant HG affected dogs exclusively expressed transcripts lacking the 34 nucleotides corresponding to the entire exon 3. In contrast, homozygous wildtype and heterozygous dogs expressed a mixture of transcripts with and without exon 3 (Fig 4).

## Validation of genotype-phenotype association

We contacted owners of four dogs with the homozygous mutant A/A genotype from our cohort of Australian Shepherd Dogs with unknown HG phenotype. One owner confirmed a HG phenotype in his dog, which had been obtained during a previous diagnostic examination. We obtained fresh blood samples from the three remaining dogs. Blood smears confirmed the HG phenotype in all three of them. The total number of homozygous mutant A/A dogs with a confirmed HG phenotype therefore increased from 26 to 30 at the end of the study.

## Discussion

In this study, we identified a homozygous *LMBR1L* splice site variant in Australian Shepherd dogs with HG and perfect genotype-phenotype association in 30 cases and 68 controls. The

mutant allele was absent from more than 900 dogs of other dog breeds. It selectively abrogates the expression of transcripts encoding the longer X2 isoform of the LMBR1L protein but does not affect the shorter X1 isoform.

*LMBR1L* encodes limb development membrane protein 1 like, which has so far not been associated with phenotypes similar to HG or PHA in humans or domestic animals. In mice, *Lmbr1l* null variants lead to aberrant activation of Wnt/β-catenin signaling, which in turn results in excessive T-cell apoptosis and lymphopenia [27]. We did not perform a detailed immunological assessment of the *LMBR1L* mutant dogs. In a previous study [23], leukocyte function did not differ in five dogs with HG/PHA compared to five control dogs. However, neither the dog breed nor the molecular cause for the observed hyposegmentation of granulocytes was reported in that study [23].

Our preliminary data do not indicate a shorter life expectancy of HG affected Australian Shepherd Dogs. The analysis of dogs that died in Switzerland between 2016 and 2020 demonstrated a median life expectancy of 11.1 years for Australian Shepherd Dogs, which is in the range of other medium-sized dog breeds [28]. In light of the global high prevalence of the HG phenotype, it seems very unlikely that a major negative impact on health would have gone unnoticed by veterinarians, breeders and owners of Australian Shepherd Dogs.

The function of the LMBR1L protein is not well characterized. Functional differences between the different LMBR1L isoforms are not known at this time. The current human annotation of the *LMBR1L* gene includes one non-coding and 17 different coding transcripts that give rise to 9 different protein isoforms. HG affected Australian Shepherd Dogs therefore provide a unique opportunity to disentangle the potential functional complexity of the different LMBR1L isoforms.

We have not been able to determine the cause of the hydranencephaly and extensive malacia in the four stillborn puppies. In retrospective, their phenotype with gross brain malformations and normal bones was quite distinct from human fetuses with Greenberg skeletal dysplasia [8] or from the reported PHA homozygous cat [16] and rabbits [13]. Therefore, the pathology and the genetic data strongly suggest that the phenotype of these puppies was unrelated to the HG phenotype. It remains unclear whether it was caused by genetic and/or environmental factors.

Our study established an autosomal recessive mode of inheritance for *LMBR1L*-related HG and refutes previous claims of a dominant inheritance for this phenotype in Australian Shepherd Dogs [24]. Our study further demonstrates that HG in Australian Shepherd Dogs is not homologous to PHA in humans as previously suggested [24]. Apparently, locus heterogeneity exists and is responsible for different forms of hyposegmentation of polymorphonuclear white blood cells. This has important consequences for breeding as the genetically distinct phenotypes may be associated with vastly different consequences for health. Further studies are urgently warranted to investigate whether *LMBR1L*-related HG in Australian Shepherd Dogs has any negative consequences for health and wellbeing. If this should not be the case, any existing restrictions in mating HG affected dogs should be lifted as such restrictions result in an unnecessary and potentially deleterious reduction of genetic diversity in the breed.

In conclusion, we have shown that HG in Australian Shepherd Dogs is an autosomal recessive trait, most likely caused by deficiency of the long isoform of the LMBR1L protein. This points to a previously unknown role of LMBR1L during the maturation and possibly in the function of myeloid cells. HG in Australian Shepherd Dogs is not homologous to human PHA, and HG affected dogs do not show any overt clinical phenotype. We propose that the use of the term PHA should be restricted to *LBR1*-related forms of HG. Our findings have major implications for the breeding of Australian Shepherd Dogs.

## Material and methods

### Ethics statement

The dogs in this study were privately owned. The post mortem examination was undertaken with owner consent, for diagnostic purposes. Samples were collected with the consent of the owners. The collection of blood samples from the dogs was approved by the Canton of Bern, permit BE71/19, and the Canton of Zurich, permit ZH225/18.

### Clinical and hematological examinations regarding the HG phenotype

Blood collection was approved by the Canton of Bern (permit BE71/19) or Canton of Zurich (permit ZH225/18). Because pathological conditions and treatment with certain medications are recognized to result in hematological changes resembling PHA (pseudo-PHA) [29,30], only healthy Australian Shepherd Dogs were included to evaluate the HG phenotype.

After obtaining written consent from the dog's owner, a general and thorough reproductive anamnesis, as well as a general physical examination were performed. Two 5 ml blood samples were collected from each animal by venipuncture into EDTA and heparin tubes, respectively. Blood examinations were performed in the Clinical Laboratory of the Vetsuisse Faculty, University of Zurich and included hematology (complete blood count, CBC) using Sysmex XT-2000iV (Sysmex Corporation, Kobe, Japan) with white blood cell (WBC) manual differential count, and a full blood biochemistry profile including the measurement of C-reactive protein (CRP) as a sensitive marker of inflammation [31] using a Cobas C 501 (Roche Diagnostics AG, Rotkreuz, Switzerland). For evaluation of the HG phenotype, a smear from each animal was prepared from fresh EDTA-anticoagulated blood and stained with Wright-Giemsa. Leukocytes were microscopically enumerated and morphologically assessed. In each dog, one hundred blood polymorphonuclear cells (granulocytes including neutrophils and eosinophils) were examined and classified by nuclear morphology (presence of unilobed (round, oval or bean-shaped) or symmetric bilobed nuclei), chromatin pattern (clumped, abnormally clumped or smooth) and characteristics of the cytoplasm (amount, color and presence of abnormal granules). The diagnosis of HG was based on microscopic examination of the blood smear showing the majority of granulocytes (mainly neutrophils and eosinophils) with hyposegmented nuclei whilst maintaining a mature chromatin pattern, in absence of toxic changes or an inflammatory leukogram. Leftover blood samples were stored at -80˚C for further analyses.

Data was recorded using Microsoft Excel and prevalence of HG was calculated using the same program. Data on all dogs in the study are compiled in the S1 Table.

### Clinical, hematological and pathological examinations in a litter with stillborn puppies

A pregnancy resulting from the mating between a HG positive bitch with a HG positive sire was examined. The dam and the sire did not share any common ancestors within three generations. The breeding was performed at the owners' decision, at which time the HG/PHA phenotype of the bitch and stud dog were unknown. Serial ultrasound examinations and CBC, blood chemistry and serum progesterone measurements of the pregnant bitch were performed from mid-gestation until close to term. After parturition, the four stillborn puppies from this litter were subjected to whole-body CT examinations (Brilliance CT, Philips, Zurich, Switzerland). The four stillborn puppies were subjected to a full post mortem examination followed by a histological examination of the main organs, bones and joints of one animal. The three surviving puppies from the same litter underwent clinical examination and blood collection for CBC,

blood chemistry with CRP and HG phenotyping at 8 weeks of age, following the protocol described above.

### DNA and RNA extraction

Genomic DNA was isolated from EDTA blood with the Maxwell RSC Whole Blood Kit, using a Maxwell RSC instrument (Promega, Dübendorf, Switzerland). Genomic DNA from liver tissue samples of the stillborn puppies was isolated with a proteinase K / phenol-extraction method [32]. Total RNA was extracted from PAXgene blood samples using the PAXgene Blood RNA Kit (PreAnalytiX GmbH, Hombrechtikon, Switzerland).

### SNV Genotyping

DNA from 22 HG affected and 54 HG unaffected Australian Shepherd Dogs was genotyped on illumina_HD canine BeadChips containing 220,853 markers (Neogen, Lincoln, NE, USA). The raw SNV genotypes are available in the S1 File. We did not have complete pedigree information on all 76 dogs that were genotyped on the SNV arrays. Some of the dogs were closely related, including, for example, the complete index family.

### Linkage, GWAS and autozygosity mapping

PLINK v.1.9 was used for basic file manipulation of the SNV genotypes [33]. Parametric linkage analysis was performed with a fully penetrant, recessive model for inheritance with the Merlin software [34].

For GWAS, we removed markers and individuals with less than 90% call rates. We further removed markers with minor allele frequency of less than 25%. An allelic GWAS was then performed with the GEMMA 0.98 software using a linear mixed model including an estimated kinship matrix as covariable to correct for the genomic inflation [35].

For autozygosity mapping, the genotype data from 22 HG affected dogs were used. Markers with missing genotypes in one of the 22 cases and markers on the sex chromosomes were excluded. The PLINK options --homozyg group, --homozyg-kb 300, --homozyg-snp 35, and --homozyg-window-snp 35 were used for the analysis. A tped-file containing the markers on chromosome 27 was visually inspected in an Excel spreadsheet to precisely delineate the homozygous shared haplotype in the cases. All reported positions correspond to the UU_Cfam_GSD_1.0 assembly.

### Whole-genome sequencing

Illumina TruSeq PCR-free DNA libraries with ~400 bp insert size were prepared from genomic DNA of one healthy dog with the HG phenotype and one of the stillborn puppies from the index litter. A total of 160 million and 212 million 2 × 150 bp paired-end reads were collected on a NovaSeq 6000 instrument corresponding to 18.2× and 22.4× coverage, respectively. Mapping and alignment were performed as described [36], and the sequence data were deposited under the study accession PRJEB16012 and sample accessions SAMEA104283461 or SAMEA6862979 at the European Nucleotide Archive (S2 Table).

### Variant calling

Variant calling was performed using GATK HaplotypeCaller [37] in gVCF mode as described [36]. To predict the functional effects of the called variants, SnpEff software together with NCBI annotation release 106 for the UU_Cfam_GSD_1.0 genome reference assembly was used [38]. For variant filtering, we used 918 genetically diverse control dog genomes (S2

Table). Variants were filtered as private to both cases, if the genotypes of control dogs were either homozygous reference or missing. None of the control genomes was derived from an Australian Shepherd Dog.

## PCR and sanger sequencing

The candidate variant *LMBR1L*:XM_038577534.1:c.191+1G>A was genotyped by direct Sanger sequencing of PCR amplicons. A 392 bp PCR product was amplified from genomic DNA using AmpliTaqGold360Mastermix (Thermo Fisher Scientific, Waltham, MA, USA) and the primers 5'-CTT TAA GCT GCC ACC TCA GC-3' and 5'-GAG TGA AGT GAA GCC GGA AC-3'. Sanger sequences were analyzed using the Sequencher 5.1 software (GeneCodes, Ann Arbor, MI, USA).

## RT-PCR for mRNA analyses

Double stranded cDNA was generated from total RNA using the SuperScript IV Reverse Transcriptase Kit (Thermo Fisher Scientific). For RT-PCR, a forward primer 5'-GGT TCG CGA GTG CAT TAT CT-3' located at the boundary of exon 1 and 2 together with a reverse primer 5'-CCA GGA CAC CCT TTC TGG AG-3' located at the boundary of exon 5 and 6 were used. After an initial denaturation of 10 min at 95˚C, 35 cycles of 30 s at 95˚C, 30 s at 60˚C, and 60 s at 72˚C were performed, followed by a final extension step of 7 min at 72˚C. The RT-PCR products were visualized using a 5200 Fragment Analyzer instrument (Agilent, Basel, Switzerland), and sequenced as described above.

## Supporting information

**S1 Fig. Re-analysis of the PHA incidence data from [24].**
(PDF)

**S1 File. SNV genotypes of 76 Australian Shepherd Dogs.**
(ZIP)

**S1 Table. Compilation of all dogs in the study.**
(XLSX)

**S2 Table. Accession numbers of 920 dog genome sequences.**
(XLSX)

**S3 Table. Raw Merlin output for the linkage analysis with respect to hydranencephaly.**
(XLSX)

**S4 Table. Summary of linkage data for hydranencephaly.**
(XLSX)

**S5 Table. Raw Plink output for homozygosity mapping of hydranencephaly.**
(XLSX)

**S6 Table. Combined linkage and homozygosity for hydranencephaly.**
(XLSX)

**S7 Table. Variant filtering in WGS data of a stillborn puppy with hydranencephaly.**
(XLSX)

**S8 Table. GWAS results for the HG locus.**
(XLSX)

**S9 Table. Combined GWAS and homozygosity for HG.**
(XLSX)

**S10 Table. Private variants in WGS data of two Australian Shepherd Dogs with HG.**
(XLSX)

## Acknowledgments

The authors would like to thank all dog owners and breeders for donating samples and information. The Australian Shepherd Club Switzerland is acknowledged for promoting the study and continued support throughout the project. Katharina Windbichler contributed to the pathological investigations of the stillborn puppies. We also wish to thank the Next Generation Sequencing Platform of the University of Bern for performing whole-genome sequencing experiments and the Interfaculty Bioinformatics Unit for providing high performance computing infrastructure. We acknowledge the DBVDC consortium, the Dog10K genomes project and all researchers who deposited dog or wolf whole genome sequencing data into public databases.

## Author Contributions

**Conceptualization:** Tosso Leeb, Orsolya Balogh.

**Data curation:** Vidhya Jagannathan.

**Funding acquisition:** Iris M. Reichler, Claude Schelling, Orsolya Balogh.

**Investigation:** Bianca Lourdes Frehner, Matthias Christen, Iris M. Reichler, Marilisa Novacco, Barbara Riond, Laureen M. Peters, José Suárez Sánchez-Andrade, Aldona Pieńkowska-Schelling, Claude Schelling, Anja Kipar, Tosso Leeb, Orsolya Balogh.

**Methodology:** Vidhya Jagannathan.

**Project administration:** Iris M. Reichler, Claude Schelling, Tosso Leeb.

**Supervision:** Iris M. Reichler, Barbara Riond, Claude Schelling, Anja Kipar, Tosso Leeb, Orsolya Balogh.

**Visualization:** Bianca Lourdes Frehner, Matthias Christen, Marilisa Novacco, Tosso Leeb.

**Writing – original draft:** Bianca Lourdes Frehner, Matthias Christen, Iris M. Reichler, Marilisa Novacco, Anja Kipar, Tosso Leeb, Orsolya Balogh.

**Writing – review & editing:** Bianca Lourdes Frehner, Matthias Christen, Iris M. Reichler, Vidhya Jagannathan, Marilisa Novacco, Barbara Riond, Laureen M. Peters, José Suárez Sánchez-Andrade, Aldona Pieńkowska-Schelling, Claude Schelling, Anja Kipar, Tosso Leeb, Orsolya Balogh.

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
