## [Decision Letter · Decision Letter 0]

7 May 2023

Dear Dr Leeb,

Thank you very much for submitting your Research Article entitled 'Autosomal recessive hyposegmentation of granulocytes in Australian Shepherd Dogs indicates a role for LMBR1L in myeloid leukocytes' to PLOS Genetics.

The manuscript was fully evaluated at the editorial level and by a single peer reviewer. Three additional reviewers declined; after discussing the manuscript and the review comments (from a reviewer who is typically very rigorous and critical) we have decided to move forward with a single review. As you will see, the reviewer is enthusiastic about the work but raises a number of minor concerns that should be straightforward to address in a revised manuscript.

We therefore ask you to modify the manuscript according to the review recommendations. Your revisions should address the specific points made by each reviewer.

Yours sincerely,

Gregory S. Barsh

Editor-in-Chief

PLOS Genetics

Gregory Copenhaver

Editor-in-Chief

PLOS Genetics

Reviewer's Responses to Questions

**Comments to the Authors:**

Reviewer #1: This manuscript describes the identification of a splice site variant in LMBR1 that is associated with hyposegmentation of granulocytes (HG) in Australian shepherd dogs (ASD). The identified variant is compelling as causal for HG because: 1) the function of the gene is consistent with the phenotype, 2) it disrupts a splice site causing complete absence of an LMBR1 isoform, 3) it segregates with the HG phenotype in a well-phenotyped cohort, and 4) it is unique to ASD. This report is important because there is limited literature about this gene.

I am intrigued by the high frequency of the variant since it appears to have occurred after breed divergence and causes an undesirable phenotype. According to GeneCards, this gene is regularly reported in GWAS for body size/weight. Maybe the allele is under selection for size in ASD, or subpopulations of ASD?

The author summary should be non-technical and appropriate for a wide audience. Parts of your author summary are more technical than the abstract (e.g., the opening sentences).

I appreciate that you are telling the story as it unfolded, but I found it a bit confusing/misleading that the summaries start by talking about PHA and Greenberg but then the story ends up not really being about either one. I also don’t like the emphasis in the summaries on how these results “refute previous hypothesis on the mode of inheritance.” I think it is fine to discuss, but the wording in the abstract and summary feels strong and distracts from the importance of the work. Additionally, the resorbed and stillborn puppies in the litter give me pause and make me wonder if, while it looks a lot like a simple recessive, maybe there IS more to it? Are there modifiers? Is it semilethal? The excessive number of stillborn puppies is what prompted the study so naturally it feels disappointing that the conclusion of the study is that the stillbirths are unrelated to the findings.

In the diagnosis of HG, is it all or none? How many abnormal cells need to be present to classify as affected?

Were the parents of the litter closely related?

The bands in Figure 1A are challenging to see.

**Have all data underlying the figures and results presented in the manuscript been provided?**

Reviewer #1: Yes

PLOS authors have the option to publish the peer review history of their article (what does this mean?). If published, this will include your full peer review and any attached files.

Reviewer #1: No

---

## [Decision Letter · Decision Letter 1]

1 Jun 2023

Dear Dr Leeb,

We are pleased to inform you that your manuscript entitled "Autosomal recessive hyposegmentation of granulocytes in Australian Shepherd Dogs indicates a role for LMBR1L in myeloid leukocytes" has been editorially accepted for publication in PLOS Genetics. Congratulations!

The revised manuscript was seen by the original reviewer who recommends acceptance.

Yours sincerely,

Gregory S. Barsh

Editor-in-Chief

PLOS Genetics

Gregory Copenhaver

Editor-in-Chief

PLOS Genetics

Comments from the reviewers (if applicable):

Reviewer's Responses to Questions

**Comments to the Authors:**

Reviewer #1: The authors have sufficiently addressed my comments.

**Have all data underlying the figures and results presented in the manuscript been provided?**

Reviewer #1: Yes

PLOS authors have the option to publish the peer review history of their article (what does this mean?). If published, this will include your full peer review and any attached files.

Reviewer #1: No

**Data Deposition**

http://datadryad.org/submit?journalID=pgenetics&manu=PGENETICS-D-23-00401R1

**Press Queries**

---

## [Editor Report · Acceptance letter]

15 Jun 2023

PGENETICS-D-23-00401R1 

Autosomal recessive hyposegmentation of granulocytes in Australian Shepherd Dogs indicates a role for LMBR1L in myeloid leukocytes 

Dear Dr Leeb, 

We are pleased to inform you that your manuscript entitled "Autosomal recessive hyposegmentation of granulocytes in Australian Shepherd Dogs indicates a role for LMBR1L in myeloid leukocytes" has been formally accepted for publication in PLOS Genetics! Your manuscript is now with our production department and you will be notified of the publication date in due course.

With kind regards,

Judit Kozma

PLOS Genetics

On behalf of:
